# In Vitro Human Monoamine Oxidase Inhibition and Human Dopamine D_4_ Receptor Antagonist Effect of Natural Flavonoids for Neuroprotection

**DOI:** 10.3390/ijms242115859

**Published:** 2023-11-01

**Authors:** Pradeep Paudel, Jae Sue Choi, Ritu Prajapati, Su Hui Seong, Se Eun Park, Woo-Chang Kang, Jong-Hoon Ryu, Hyun Ah Jung

**Affiliations:** 1Invasive Insect Biocontrol and Behavior Laboratory, Beltsville Agricultural Research Center-West, USDA-ARS, Beltsville, MD 20705, USA; 2Department of Food and Life Science, Pukyong National University, Busan 48513, Republic of Korea; choijs@pknu.ac.kr (J.S.C.); ritpraz@gmail.com (R.P.); shseong@hnibr.re.kr (S.H.S.); gogo1685@naver.com (S.E.P.); 3Natural Products Research Division, Honam National Institute of Biological Resource, Mokpo 58762, Republic of Korea; 4Department of Oriental Pharmaceutical Science, College of Pharmacy, Kyung Hee University, Seoul 02447, Republic of Korea; jokerharry@khu.ac.kr (W.-C.K.); jhryu63@khu.ac.kr (J.-H.R.); 5Department of Food Science and Human Nutrition, Jeonbuk National University, Jeonju 54896, Republic of Korea

**Keywords:** flavonoids, hMAO, GPCRs, dopamine, antagonist, serotonin, neuroprotection

## Abstract

Natural flavone and isoflavone analogs such as 3′,4′,7-trihydroxyflavone (**1**), 3′,4′,7-trihydroxyisoflavone (**2**), and calycosin (**3**) possess significant neuroprotective activity in Alzheimer’s and Parkinson’s disease. This study highlights the in vitro human monoamine oxidase (hMAO) inhibitory potential and functional effect of those natural flavonoids at dopamine and serotonin receptors for their possible role in neuroprotection. In vitro hMAO inhibition and enzyme kinetics studies were performed using a chemiluminescent assay. The functional effect of three natural flavonoids on dopamine and serotonin receptors was tested via cell-based functional assays followed by a molecular docking simulation to predict interactions between a compound and the binding site of the target protein. A forced swimming test was performed in the male C57BL/6 mouse model. Results of in vitro chemiluminescent assays and enzyme kinetics depicted **1** as a competitive inhibitor of hMAO-A with promising potency (IC_50_ value: 7.57 ± 0.14 μM) and **3** as a competitive inhibitor of hMAO-B with an IC_50_ value of 7.19 ± 0.32 μM. Likewise, GPCR functional assays in transfected cells showed **1** as a good hD_4_R antagonist. In docking analysis, these active flavonoids interacted with a determinant-interacting residue via hydrophilic and hydrophobic interactions, with low docking scores comparable to reference ligands. The post-oral administration of **1** to male C57BL/6 mice did not reduce the immobility time in the forced swimming test. The results of this study suggest that **1** and **3** may serve as effective regulators of the aminergic system via hMAO inhibition and the hD_4_R antagonist effect, respectively, for neuroprotection. The route of administration should be considered.

## 1. Introduction

According to the United States Centers for Disease Control and Prevention, more than 200 million people, equivalent to roughly 60% of the population, suffer from at least one neurological disorder in the USA. Neurological disorders vary in severity and range from tension-type headaches and anxiety to strokes and dementia. Among these 200 million, more than 3.4 million have epilepsy, 6.2 million have Alzheimer’s or related dementias, and 40 million have anxiety disorders. Furthermore, major depressive disorder is the leading cause of disability for people between 15 and 44, and 19% of adults reported experiencing mental illness in 2019. The causes of these diseases vary, and in some cases, they are not completely known because they are multifactorial.

Consequently, treatment is very challenging. The rate at which people are seeking treatment for mental health issues has increased as well. In 2020, 93% more people took a screening test for anxiety disorders than in 2019, and there was a 62% increase in depression screenings for the same period [1]. Given the extent to which these disorders proliferate, the need for novel therapeutic strategies to manage them is urgent.

Tryptophan (TRP) is the most prevalent amino acid, which plays a significant role in protein biosynthesis, and its metabolites are implicated in central nervous system (CNS) disorders [2,3]. This amino acid metabolizes to 5-hydroxytryptophan (5-HT) and kynurenine (Kyn) via two pathways: the serotonin and kynurenine pathways, respectively [4,5]. Tryptophan hydroxylase 1 or 2 converts tryptophan to 5-hydroxytryptophan (5-HTP), the first rate-limiting step in the 5-HT pathway. 5-HTP is then decarboxylated by aromatic acid decarboxylase (AADC) to form 5-HT, which is metabolized by aralkylamine *N*-acetyltransferase (AANAT) to *N*-acetyl serotonin (NAS) and then by *N*-acetylserotonin *O*-methyltransferase (ASMT) to form melatonin. 5-HT is also metabolized by monoamine oxidase (MAO) to form 5-hydroxyindoleacetic acid (5-HIAA), which is the main metabolite of 5-HT [6]. Similarly, Trp is converted to kynurenine (Kyn) via indoleamine 2-3-dioxygenases 1 and 2 and tryptophan 2,3-dioxygenase of the Kyn pathway. Kyn is then converted to 3-hydroxyKyn (3-HK) via kynurenine 3-monooxygenase. Kynurenine aminotransferase (KAT) then converts 3-HK to xanthurenic acid (XA), which is converted to 3-hydroxyanthranilic acid (3-HAA) by kynureninase (Kynu). The 3-HAA is metabolized to picolinic acid (PA) via aminocarboxymuconate-semialdehyde decarboxylase (ACMSD) and nonenzymatically to quinolinic acid (QA). QA is then metabolized to NAD^+^ by quinolinate phosphoribosyl transferase (QPRT) [7]. These tryptophan metabolites play important roles in many psychiatric disorders, including depression [8,9].

TRP and its metabolites play an important role in alleviating various diseases ranging from psychiatric/neurological disorders to cancer [10]. This distinctive feature makes the study of TRP metabolism an exciting area of research for biomedical researchers focused on developing and identifying new therapeutic targets. Recent research suggests that the kynurenine pathway can improve many biological systems that function poorly in psychiatric disorders, including the neurotransmitters of the CNS and the immune-inflammatory system [11,12]. The kynurenine pathway event-targeted process represents an excellent opportunity to develop effective treatments for neuropsychiatric disorders [2,13].

Different classes of antidepressant drugs are available on the market for the treatment of depression with different mechanisms of action, including selective serotonin and norepinephrine reuptake inhibitors (Duloxetine, Venlafaxine, and Levomilancipran), tricyclic antidepressants (Doxepin, Imipramine, Nortriptyline, and Trimipramine), serotonin reuptake inhibitors (Sertraline, Fluoxetine, Citalopram, and Paroxetine), and monoamine oxidase inhibitors (MAOIs) (Phenelzine, Selegiline, and Tranylcypromine) [14]. However, these drugs have some side effects. Drugs used to modulate serotonin levels, such as serotonin reuptake inhibitors, can also cause serotonin toxicity (i.e., serotonin syndrome) due to their irreversible properties. Likewise, non-selective and irreversible MAO inhibitors can cause the cheese effect, hypertension, etc. Dizziness, sexual dysfunction, dry mouth, slowness of movement, and weight gain [15,16,17] are other side effects of traditional antidepressants.

Brain neurotransmitter levels are regulated by MAO-A (serotonin and norepinephrine) and MAO-B (phenethylamine) or both (dopamine) as preferred substrates [18]. Tranylcypromine and Phenelzine are the classic MAOIs (irreversible and nonselective), and moclobemide (reversible and selective) is commonly used to treat depression. Selegiline and rasagiline are selective MAO-B inhibitors that increase dopamine in the basal ganglia for the treatment of Parkinson’s disease. MAO-A inhibitors increase serotonin and norepinephrine in nerve endings and are thought to reverse the “monoamine deficiency”. Therefore, MAOIs are the oldest class of antidepressants and are effective in treating patients with unipolar major depressive disorder (MDD) and certain anxiety disorders. Although MAOIs are highly effective, clinical use is limited by declining clinical experience and a lack of understanding of dietary restrictions and drug interactions, and most psychiatrists rarely prescribe them. However, given the high rates of drug resistance in patients with major depressive disorder, the clinical efficacy of selected irreversible MAOIs for treatment-resistant depression, and changes in technology, there has been renewed interest in the clinical application of MAOIs [19].

Natural flavonoids have a long history of medical use for the treatment of various medical ailments, including inflammation [20], cancer [21,22,23], microbial infection [24], diabetes, and obesity [25]. Isoflavonols such as genistein, glycetein, calcycosin, and 3′,4′,7-trihydroxy isoflavone as well as coumestans such as coumestrol are the dietary phytoestrogens occurring in soybeans and other medicinal plants such as *Pueraria lobata* [26,27]. These phytoestrogens have shown promising therapeutic effects like anti-cancer [28,29], anti-melanogenesis [30], anti-diabetic [31,32], anti-obesity [33,34], anti-oxidant [35], anti-inflammatory [35], neuroprotective [36,37] and anti-depressive [32,38]. Modern research on natural flavonoids has revealed that dietary consumption of flavonoids and flavonoid-rich foods significantly improves cognition and delays age-related neurodegenerative disorders via inhibition of cholinesterase and β-secretase (BACE1), antioxidant mechanisms and modulation of signaling pathways that are implicated in cognitive and neuroprotective functions [39,40]. What remains lacking, however, is knowledge of the potential role of natural flavonoids in regulating aminergic pathways. There is, therefore, a critical need to define the therapeutic efficacy of natural flavonoids in depression-like animal models. Without such information, the promise of natural flavonoids for the treatment of neurodegenerative diseases will likely remain limited.

Our long-term goal is to discover natural flavonoids as hMAO inhibitors and hD_4_R antagonists for the management of neurodegenerative diseases, particularly depression. The overall objectives, which are the next step toward the attainment of our long-term goal, are to (i) elucidate the mechanism(s) of hMAO inhibition via in vitro enzyme assays and in silico molecular docking; (ii) evaluate modulating effects on dopamine (DA) and serotonin (5-HT) receptors via GPCR-functional assays and in silico molecular docking; and (iii) determine their in vivo anti-depressant efficacy using depression-like animal models. Our central hypothesis is that natural flavonoids with in vitro hMAO inhibition and hD4R antagonism have in vivo anti-depressant effects. Therefore, to attain the overall objectives and test the hypothesis, we emphasized the “One-compound multiple-targets paradigm” and evaluated the role of three naturally occurring flavonoids, namely, 3′,4′,7-trihydroxyflavone (**1**), 3′,4′,7-trihydroxyisoflavone (**2**), and calycosin (**3**), in hMAO inhibition and GPCR modulation, followed by an in vivo anti-depressant effect in depression-like animal models.

## 2. Results

### 2.1. Human Monoamine Oxidase Inhibition

The human monoamine oxidase inhibitory activity of the naturally occurring flavonoids **1**–**3** (Figure 1) was tested using recombinant hMAO-A and hMAO-B isozymes with non-specific MAO-A and -B substrates. Table 1 tabulates the in vitro hMAO inhibition potential of flavonoids **1**–**3** along with reference inhibitors. As shown in Table 1, 3′,4′,7-trihydroxyflavone (**1**) showed the most promising inhibition of hMAO-A with an IC_50_ value of 7.57 ± 0.14 µM.

Calycosin (**3**) and 3′,4′,7-trihydroxyisoflavone (**2**) showed moderate inhibition of hMAO-A with IC_50_ values of 113.78 ± 3.39 and 176.79 ± 7.80 µM, respectively. The human MAO-A-specific reference inhibitor clorgyline-HCl had an IC_50_ value of 0.02 ± 0.00 µM.

Likewise, for hMAO-B inhibition, calycosin (**3**) was the most potent among the tested flavonoids, with an IC_50_ value of 7.19 ± 0.32 µM. Flavonoid **2** showed moderate hMAO-B inhibition with an IC_50_ value of approx. 70 µM. Interestingly, flavonoid **1** was much less active at hMAO-B inhibition, depicting its selectivity towards hMAO-A. The hMAO-B selective reference inhibitor safinamide mesylate showed an IC_50_ value of 0.23 ± 0.01 µM.

Overall, the in vitro enzyme assay depicted the selectivity of **1** and **3** towards hMAO-A and hMAO-B, respectively. Based on the IC_50_ values and their selectivity, flavonoids **1** and **3** could be of interest in neurodegenerative diseases.

To better understand the enzyme inhibition mode of the most potent inhibitors (**1** for hMAO-A and **3** for hMAO-B), a kinetic study was conducted in a similar way to enzyme inhibition but at different substrate concentrations (Figure 2 and Figure 3 and Table 1). From the Lineweaver-Burk plots in Figure 2A and Figure 3A, increasing concentrations of flavonoids resulted in constant *V_max_* and increased *K*_m_ values. These results demonstrate that the flavonoids are competitive inhibitors of hMAO-A (flavonoid **1**) and hMAO-B (flavonoid **3**) with *K*_i_ values of 2.03 and 2.56 µM, respectively.

### 2.2. Computational Analysis of Human Monoamine Oxidase Inhibition

From the in vitro enzyme inhibition result, it can be seen that a small change in parent structure affected the inhibition potency and selectivity over two isoforms. Therefore, to explain the variation in potency and selectivity and gain insights into enzyme-inhibitor interactions, computational docking simulation of **1** and **2** binding to the hMAO-A (Figure 4) and **2** and **3** binding to the hMAO-B (Figure 5) was conducted via AutoDock 4.2 using the crystal structure of hMAO-A in complex with reversible inhibitor harmine (PDB code: 2Z5X) and hMAO-B in complex with reversible inhibitors safinamide (PDB code: 2V5Z). The computational docking results of the test ligands revealed that they all bind to the enzyme with high binding affinity and low binding energy. The binding energies and interacting residues of the investigated ligands, along with the reference ligands, are shown in Table 2 and Table 3. Among the tested flavonoids, **1** and **2** displayed good binding affinity to hMAO-A, which was comparable to the binding of the selective hMAO-A reference inhibitor harmine (−8.43 kcal/mol). Flavonoids **1** and **2** formed H-bond interactions with Asn181 and Phe208. Additionally, H-bond interactions with Gln215 and Gly443 were also observed for flavonoid **2**−hMAO-A binding. Likewise, interactions with FAD and Phe352 (Pi–Pi T-shaped), Tyr407 and Tyr444 (Pi–Pi Stacked), Ile335 and Leu337 (Pi–Alkyl) were commonly observed for **1** and **2** binding with the enzyme (Figure 4).

As tabulated in Table 3, **2** and **3** bound competitively to the catalytic site of hMAO-B with the best pose, as indicated by their low binding score, which was better than the reference inhibitor safinamide. The lowest binding score was predicted for the most potent compound, **3** (−9.99 kcal/mol), followed by **2** (−9.56 kcal/mol). Those test ligands formed H-bond interactions with Cys172 and Ile199. In addition to this, hydrophobic interactions with FAD and tyrosine residues (Tyr326, Tyr398, and Tyr435) were observed for **2**, **3**, and safinamide binding.

### 2.3. 3′,4′,7-Trihydroxyflavone as Dopamine D_4.4_R Antagonist

Test compounds **1**–**3** were first screened for functional effect over human dopamine receptors (subtypes D_1_R, D_2L_R, D_3_R, and D_4.4_R) and serotonin 5HT_1A_R at a 100 μM concentration. The screening result is tabulated in Table 4 and Figure 6. As shown there, only **1** showed significant functional effects at the test concentration on hD_4.4_R. Flavonoid **1** inhibited the response of 100 µM of dopamine by 134.47 ± 25.54%. Based on the screening result, the concentration-dependent response was tested for flavonoid **1**. As shown in Figure 6A, flavonoid **1** inhibited the response of the control agonist by 40.9, 57.65, 68.95, and 134.47% at 12.5, 25, 50, and 100 µM, respectively, yielding an IC_50_ value of 22.47 ± 2.18 µM.

The underlying mechanism of flavonoid **1**−hD_4.4_R binding was predicted via molecular docking simulation using AutoDock 4.2 (Figure 7) and compared with the reference ligands clorgyline and nemonapride (Figure 8 and Figure 9). Overall docking results, including the binding score and list of interacting amino acid residues, are tabulated in Table 5. As tabulated there, flavonoid **1** interacted with the crystal structure of hD_4.4_R (5WIU) with a low binding score (−7.71 kcal/mol) that was comparable to the reference ligand clorgyline (−8.67 kcal/mol), involving H-bond interactions with Val193 (helix V), Leu187 (ECL_2_), Ser197, and Ser196 (helix V) and hydrophobic interactions with His414 (helix VI), Leu187 (ECL_2_), Val193 (helix V), Arg186, and Val116 (helix III).

### 2.4. Behavior of Mice in Forced Swim Test

In the forced swim test, we found no significant differences in the total number of immobility episodes between the groups (Figure 10). As shown in Figure 10, the immobility time in the control and test groups is similar. Compound **1** was tested at a dose of 3, 10, and 30 mg/kg p.o., and none of those doses alleviated the immobility time. Only desipramine at a 15 mg/kg dose p.o. showed a significant reduction in immobility time. Despite being a potent hMAO inhibitor (Table 1) and hD4.4R antagonist, **1** did not show an effective reduction in immobility time in the forced swimming test (Figure 10).

## 3. Discussion

Monoamine oxidases (MAOs) are the mammalian flavoproteins that play important roles in the deactivation of biological amines such as dopamine, serotonin, norepinephrine, epinephrine, phenylethylamine, benzylamine, melatonin, tyramine, and tryptamine [41]. Catalytic degradation of biological amines via MAOs increases biogenic amine turnover, leading to greater oxidative stress and the subsequent acceleration of the neurodegenerative process [42]. Thus, MAOs are the known targets for many neurodegenerative disorders and depressive illnesses. The levels of serotonin and nor-epinephrine are regulated by MAO-A in the brain, and thus, MAO-A inhibitors such as tranylcypromine and Phenelzine (irreversible and non-selective MAO-inhibitors) and moclobemide (reversible and selective MAO-A inhibitors) are used for treating depression [43]. Of the two isoforms, MAO-B predominates in the brain and is responsible for lowering the levels of dopamine and phenylethylamine. Selective MAO-B inhibitors such as selegiline and rasagiline enhance the dopamine level in the basal ganglia and are clinically approved for PD treatment [43,44]. Recent studies suggest a close correlation between MAO activity and the progression of AD, characterized by elevated MAO activity around Aβ plaques [45] and enhanced Aβ production in neurons with increased MAO-B levels [46]. Further, siRNA silencing of MAO-B in the primary cortical neurons considerably lowered intracellular Aβ levels [46]. Also, multiple pharmacological studies have demonstrated the neuroprotective role of MAO inhibitors in the prevention and treatment of AD [47].

Natural products based on flavonoids–scaffolds possess different biological and pharmacological properties such as antioxidant, anti-inflammatory, neuroprotective, acetylcholinesterase (AChE), and butyrylcholinesterase (BChE) inhibition, anti-Aβ fibril formation, β-secretase, and MAO inhibition [48]. Also, recent studies demonstrate that flavonoids modulate CNS GPCRs such as dopamine, serotonin, and vasopressin receptors. For instance, Park et al. (2020) identified luteolin, 3′,4′,5,7-tetrahydroxyflavone, as dopamine D_4_ and vasopressin V_1A_ receptor antagonists. Additionally, they found that it is an MAO-A inhibitor with an IC_50_ value of 8.57 ± 0.47 µM [49]. Likewise, a prenylated flavanone, kurarinone, was also found to be a D_1A_R antagonist and a D_2L_ and D_4_ agonist [50]. 3′,4′,7-Trihydroxyflavone (**1**) decreases NO production and exerts an anti-neuroinflammatory effect through the inhibition of the JNK-STAT1 pathway in microglia [51]. Further, **1** protects the neuronal cells from H_2_O_2_-induced oxidative stress and cytotoxicity [52]. From different studies, calycosin (**3**), a 3′,7-dihydroxy-4′methoxyisoflavone, has exhibited neuroprotective effects mediated through its anti-oxidant and anti-inflammatory actions [53,54,55]. An in vitro and in vivo study by Yang et al. (2018) revealed that **3** attenuates the MPTP-induced inflammatory response and alleviates dyskinesia and sensory disturbances in MPTP-induced PD mice, thus suggesting its potential use against PD [55]. Also, Oh et al. (2020) identified it as a selective hMAO-B inhibitor [56]. Coumestrol is another isoflavonoid phytoestrogen whose protective actions against neonatal hypoxia-ischemia [57], cerebral ischemia [58], and amyloid beta-induced and LPS-induced toxicity on mouse astrocytes have been established [59]. Our previous study demonstrated the anti-Alzheimer’s, anti-oxidant, and anti-depressant properties of coumestrol [60,61]. These findings led us to the investigation of compounds **1** and **3,** along with **2**, an isoflavone analog of **1,** against hMAO, since the hMAO inhibition potential of **1** and **2** is yet to be explored. Given the evidence that the flavonoids can modulate dopamine and serotonin receptors [62], we used GPCR functional assays to test these compounds for their functional activity.

In the hMAO-A inhibition assay, flavonoid **1** showed significant inhibition of hMAO-A with an IC_50_ value of 7.57 ± 0.14 µM. While the isoflavone analogs **2** and **3** showed only a modest inhibition against hMAO-A. However, in the case of the hMAO-B inhibition assay, **3** displayed a notable inhibition with an IC_50_ value of 7.19 ± 0.32 µM. Compound **2** showed moderate inhibition of hMAO-A with IC_50_ values of 71.23 ± 0.06 µM. From a structural point of view, calycosin (**3**) is an O-methylated isoflavone, and **2** is its structural analog. The structural difference between **2** and **3** is that the 4′-methoxy group in **3** is replaced with a hydroxy group in **2**. Both showed mild inhibition of hMAO-A, but the effect is good at hMAO-B. Interestingly, **3** showed a ten-times more potent effect (IC_50_: 7.19 ± 0.32 µM) than **2** (IC_50_: 71.23 ± 0.06 µM). Comparing the results, we can assume that the 4′-methyl substitution in the isoflavone ring increases the hMAO-B inhibition effect. MAO inhibition is strongly dependent on the presence of a (*p*-OH-substituted) phenyl at C2, unsaturation at the C2–C3 positions of the structure, the possibility of establishing hydrophobic interactions, and ring planarity [63]. Corroborating with the earlier study, compound **3** showed selective hMAO-B inhibition [56]. However, IC_50_ values vary, which may be due to different experimental conditions. The enzyme kinetic study of **1** and **3** on hMAO-A and hMAO-B revealed the competitive inhibition mode. Interactions with this aromatic cage-forming covalent FAD coenzyme and tyrosine residues of the active site of hMAO-A are considered important for catalytic activity [64]. Compounds **1** and **2** have many common hydrophobic interactions with hMAO-A active site residues such as FAD, Tyr407, Ile335, and Ile180; however, additional interaction with Phe208 via Pi–Pi T-shaped bond might be responsible for its specificity towards hMAO-A and higher potency. Phe208 and Ile335 are critical residues for inhibitor selectivity [65]. Considering the inhibition of hMAO-A and hD_4.4_R antagonism by **1**, we conducted the forced swimming test to evaluate its anti-depressive activities. Unfortunately, we did not observe any positive effects (Figure 10). Our data are surprising, as previous studies have shown treatment with oral antidepressants may reduce the immobility time of Swiss mice in the Porsolt test in various experimental conditions [66,67]. As discussed earlier [68], the lack of efficacy of oral administration could be the reason behind the ineffectiveness of **1** in reducing immobility time to show an anti-depressant effect.

In the GPCR functional assay, only **1** showed significant functional effects at the test concentration on hD_4.4_R. Flavonoid **1** inhibited the response of the control agonist at hD_4.4_R with an IC_50_ value of 22.47 ± 2.18 µM. The 4-oxo group in the C-ring of flavonoid **1** formed an H-bond interaction with Leu187, and the hydroxyl group at position C-3′ in the B-ring formed H-bond interactions with Ser196, Ser197, and Val193. The dopamine D4 receptor (D_4_R) plays important roles in cognition, attention, and decision making, and pharmacological activation of D_4_R could be useful in treating cognitive deficits associated with schizophrenia [69,70,71,72] and attention-deficit/hyperactivity disorder [72,73]. While D_4_R antagonism may be useful in treating substance use disorder [74] and L-DOPA-induced dyskinesias (LID) [75].

The etiology of LID is unknown. Previous studies have suggested that LID may be associated with L-DOPA-induced increases in synaptic DA levels in the caudate putamen [76,77]. In addition, D_4_Rs are found in the basal ganglia region, which is a key region for dyskinesia [78]. So D_4_R antagonists could fill the gap in LID attenuation in PD patients [79]. Likewise, the most recent study has shown that D_4_R antagonists selectively inhibit the growth of glioblastoma neural stem cells [80]. Therefore, novel D_4_R-selective ligands have promise for developing medications for neuropsychiatric conditions. This study highlights the antagonistic effect of **1** on D_4_R. To our knowledge, this is the first study to report this specific effect of this compound.

### Limitations and Future Directions

Our study highlights the in vitro human monoamine oxidase inhibition and modulating effect on dopamine (DA) and serotonin (5-HT) receptors via GPCR-functional assays and predicts the binding mode. In silico molecular dynamics studies predicting the stability of ligand–receptor complexes are lacking. Penetration into the central nervous system and the stability of the ligand–receptor complex remain to be studied in vivo. The small number of test compounds limited the structure–activity relationship. The effects of the tested flavonoids at the cellular or organismal level remain unknown because in silico modeling cannot account for interactions between compounds and other unrelated targets. Also, other routes of administration should be considered, as the oral route might not be effective for anti-depressant effects.

In the future, a more detailed understanding of hD_4.4_R signaling and regulation, especially using in vivo models, will be critical to ensuring the activity of these natural flavonoids, particularly **1** in neurogenerative diseases. The anti-depressant effect of **1** should be evaluated from different routes of administration to conclude our hypothesis—natural flavonoids with in vitro hMAO inhibition and hD_4.4_R antagonism have in vivo anti-depressant effects.

## 4. Materials and Methods

### 4.1. Flavonoids, Chemicals, and Reagents

Flavonoids **1** and **2** were obtained from Indofine Chemical Company, Inc. (Hillsborough, NJ, USA). Flavonoid **3** was isolated from the *n*-hexane fraction of *Pueraria lobata* root. The purity of the isolated compounds was considered to be >98%, as evidenced by spectral data [31]. An MAO-GloTM assay kit was purchased from Promega (Promega Corporation, Madison, WI, USA). Transfected CHO and Ba/F3 cells were obtained from Eurofins Scientific (Le Bois I’Eveque, France). Hank’s balanced salt solution (HBSS), Dulbecco’s modified Eagle medium, and 4-(2-hydroxyethyl)-1-piperazineethanesulfonic acid (HEPES) buffers were obtained from Invitrogen (Carlsbad, CA, USA). The hMAO isozymes and reference drugs clorgyline-HCl, safinamide mesylate salt, l-deprenyl HCl, dopamine, (S)-WAY-100635, serotonin, and butaclamol were purchased from Sigma-Aldrich (St. Louis, MO, USA). All chemicals and solvents used for column chromatography were of reagent grade and purchased from commercial sources.

### 4.2. In Vitro Human MAO Inhibition and Enzyme Kinetics

A chemiluminescent assay was performed in a white, opaque 96-well plate using the MAO-Glo kit (Promega, Madison, WI, USA) to evaluate human monoamine oxidase (hMAO) inhibitory potential. Detailed experimental conditions and procedures were reported previously [81,82]. The test compounds were evaluated at different concentrations to obtain an IC_50_ value. The percent of inhibition (%) was obtained with the following equation: % Inhibition = *(A_c_* − *A_s_)/A_c_* × 100, where *A_c_* is the absorbance of the control and *A_s_* is the absorbance of the sample. For hMAO-A, clorgyline was used as a positive control. Whereas for hMAO-B, safinamide mesylate was a reference compound. l-Deprenyl HCl was also used as a positive control for both hMAO-A and hMAO-B inhibition assays.

The kinetic analysis of hMAO inhibition was analyzed at different concentrations of hMAO substrate depending on the isozyme (40, 80, and 160 µM for hMAO-A and 4, 8, and 16 µM for hMAO-B) following the same method of enzyme inhibition. The concentrations of the test compounds for the kinetic study are presented in Figure 2 and Figure 3. Kinetic parameters were analyzed using SigmaPlot (v12.0, SPP Inc., Chicago, IL, USA).

### 4.3. Cell-Based Functional GPCR Assay

Cell-based functional GPCR assays were conducted in CHO cells and Ba/F3 cells transfected with a plasmid containing the GPCR gene of interest. The functional activity of the test compounds (agonist or antagonist) was evaluated by measuring their effects on cAMP modulation or Ca^2+^ ion mobilization, depending on the receptor type. All assays were performed at Eurofins Cerep (Le Bois I’Eveque, France) following their in-house protocol, as stated in our previous reports [83,84,85].

#### 4.3.1. Measurement of cAMP Level

The functional activity of the test compounds on D_1_R, D_3_R, and D_4_R was assessed by evaluating the effect on cAMP modulation. For this, stable transfectants (CHO-D_1_R, CHO-D_3_R, and CHO-D_4_R) were suspended in HBSS (Invitrogen, Carlsbad, CA, USA) containing 20 mM of HEPES buffer and 500 μM of 3-isobutyl-1-methylxanthine, distributed into microplates (5 × 10^3^ cells/well), and incubated for 30 min at room temperature (RT) in the absence (control) or presence of the test compounds (6.25, 12.5, 25, 50, and 100 μM) or reference agonist (DA). In the D_3_R and D_4_R assays, the adenylyl cyclase activator NKH 477 was added at a final concentration of 1.5 and 0.7 μM and incubated for 30 and 10 min, respectively, at 37 °C. Then, the cells were lysed, and a fluorescence acceptor (D2-labeled cAMP) and fluorescence donor (an anti-cAMP antibody with europium cryptate) were added. The fluorescence transfer was measured at λ_ex_ = 337 nm and λ_em_ = 620 and 665 nm using a microplate reader (Envision, Perkin Elmer, Waltham, MA, USA) after 60 min of incubation at RT. Agonist effects are expressed as the % of the control response to 10 μM of DA for D_1_R and 300 nM of DA for D_3_R/D_4_R. Similarly, antagonist effects are expressed as the % inhibition of the control response to DA at 300 nM for D_1_R, 10 nM for D_3_R, and 100 nM for D_4_R. The reference agonist DA and antagonists SCH 23390, (+)-butaclamol, and clozapine were used to validate the study.

#### 4.3.2. Measurement of Intracellular [Ca^2+^] Levels

The functional activity of the test compounds on D_2_R and 5HT_1A_R was tested by fluorimetrically evaluating their effect on cytosolic Ca^2+^ ion mobilization. In brief, CHO-D_2_R and Ba/F3-5HT_1A_R cells were separately suspended in HBSS (Invitrogen, Carlsbad, CA, USA) complemented with 20 mM HEPES buffer and distributed into microplates (1 × 10^5^ cells/well). Then, a fluorescent probe (Fluo8, AAT Bioquest, Sunnyvale, CA, USA) mixed with probenecid in HBSS (Invitrogen, Carlsbad, CA, USA) supplemented with 20 M HEPES (Invitrogen) (pH 7.4) was added to each well, and the cells were allowed to equilibrate for 60 min at 37 °C. Thereafter, the plates were positioned in a microplate reader (FlipR Tetra, Molecular Device), and compounds **1**–**3** (6.25, 12.5, 25, 50, and 100 μM), reference agonist, or HBSS (basal control) were added. We then measured the fluorescent intensity, which varied in proportion to the free cytosolic Ca^2+^ ion concentration. Agonist effects are expressed as the % of the control response to 10 μM DA for D_2_R and 2.5 μM serotonin for 5HT_1A_R. Similarly, antagonist effects are expressed as the % inhibition of the control response to 700 nM DA for D_2_R and 30 nM serotonin for 5HT_1A_R. Reference agonists (DA and serotonin) and antagonists (butaclamol and (S)-WAY-100635) were used to validate the study.

### 4.4. In Silico Molecular Docking Simulation

Automated single docking simulations were carried out with AutoDock 4.2 [86]. X-ray crystallographic structures of hMAO-A and hMAO-B were obtained from the PDB with IDs 2BXR and 2BYB. The 3D chemical structures of four test compounds were obtained from PubChem Compound (NCBI, CIDs 5,322,065, 5,284,648, and 5,280,448 for compounds **1**–**3**, respectively). The crystal structures of the reference compounds safinamide, harmine, nemonapride, and clorgyline were also obtained from NCBI under CIDs 131,682, 5,280,953, 156,333, and 4380, respectively. Water and ligand molecules were removed using Discovery Studio (v17.2, Accelrys, San Diego, CA, USA). In the case of the human MAO isozymes, the co-factor flavin adenine dinucleotide (FAD) was retained. X-ray crystallographic structures with PBD IDs 5WIU (the best resolution structure of hD4R) were used for the hD4 receptor [87]. The Lamarckian genetic algorithm method in AutoDock 4.2 was applied. For the docking calculations, Gasteiger charges were added by default, and all the torsions were allowed to rotate. The grid maps were generated with the AutoGrid program. The docking protocol for rigid and flexible ligand docking consisted of 10 independent genetic algorithms, and other parameters were set using the defaults in the AutoDock Tools. The docking results were visualized using Discovery Studio.

### 4.5. Animal

Male C57BL/6 mice (22–26 g, 7 weeks) were purchased from Orient Bio Inc. (Seongnam-si, Korea), a branch of Charles River Laboratories (Seoul, Republic of Korea), and kept in the University Animal Care Unit for 1 week before the experiments. Five animals were housed per cage and allowed access to water and food ad libitum; the environment was maintained at a constant temperature (23 ± 1 °C) and humidity (60 ± 10%) under a 12 h light/dark cycle (the lights were on from 07:30 to 19:30 h). The treatment and maintenance of the animals were carried out according to the Animal Care and Use Guidelines of Dong-A University, Republic of Korea. All in vivo experiments were performed according to the protocols approved by the Institutional Animal Care and Use Committee of Dong-A University (approved protocol numbers: DIACUC-approved-17-20) and were per the National Institutes of Health guidelines.

### 4.6. Forced Swim Test

A forced swim test (FST) was performed in a transparent cylindrical glass cylinder with a diameter of 14 cm and a height of 25 cm filled with 20 cm of water (24 ± 2 °C). Mice were habituated to the experimental environment 1 h before the test. The 3′,4′,7-trihydroxyflavone (3, 10, or 30 mg/kg, p.o.) or the same volume of vehicle was administered 1 h before FST. As a positive control, desipramine (15 mg/kg, i.p.) was administered 30 min before FST. Mice were immersed in water and recorded with a video camera for a total of 6 min. Immobility time was analyzed for the last 4 min of the 6 min using a video-based Ethovision System (Noldus, Wageningen, The Netherlands).

### 4.7. Statistical Analysis

Statistical analysis was performed with One-way ANOVA and Student’s *t*-test using Microsoft Excel 2016 (Microsoft Corporation, Redmond, WA, USA). All in vitro experiments were carried out in triplicate on three individual days and are expressed as the mean ± standard deviation (SD). Differences were considered statistically significant at a *p*-value < 0.01 compared to the vehicle-treated control group.

## 5. Conclusions

Our study is the first to assess the dopamine (DA) and serotonin (5-HT) receptors modulating activity and hMAO inhibitory potency of natural flavonoids **1**–**3**. Little is known about the pharmacological importance of these flavonoids in regulating GCPR. GPCR functional screening revealed that **1** possesses hD_4.4_R antagonist properties. Likewise, **1** showed promising inhibition of hMAO-A, and **3** was the most potent inhibitor of hMAO-B. Due to the multifactorial complexities associated with NDDs such as AD and PD, molecules with multitargeting properties and minimal toxicity hold promise as potential therapeutic approaches. Therefore, the versatility of **1** may help target different root causes of NDD and improve related symptoms. A recent study confirmed that intravenous injections of cannabidiol can induce long-term antidepressant-like effects [88]. The findings indicated that 10 mg/kg cannabidiol (i.v.) or 100 mg/kg CBD (p.o.) administered weekly for four weeks can significantly improve depressive symptoms. In contrast, no significant effects were observed at 10 mg/kg cannabidiol p.o. In our study, oral administration of **1** in the male C57BL/6 mice model at a dose of up to 30 mg/kg did not show an antidepressant effect. So, the effect should be evaluated at different routes of administration or at higher doses following p.o. administration. Further, in vivo studies are needed to support the regulatory role of DAR in vitro and to observe the efficacy of **1** in alleviating DAergic neurodegeneration and disease conditions in animal models.

## Figures and Tables

**Figure 1 ijms-24-15859-f001:**
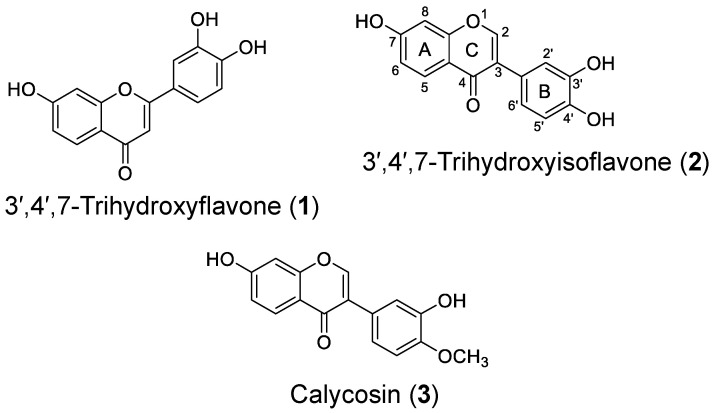
Chemical structures of test flavonoids.

**Figure 2 ijms-24-15859-f002:**
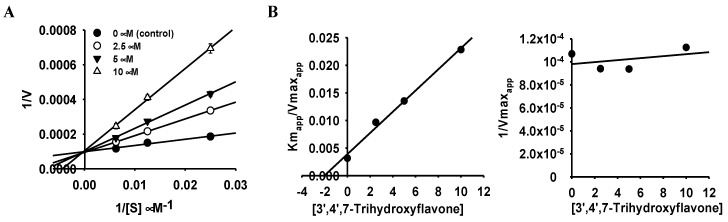
Lineweaver–Burk plots (**A**) and secondary plots (**B**) of 3′,4′,7-trihydroxyflavone for hMAO-A inhibition.

**Figure 3 ijms-24-15859-f003:**
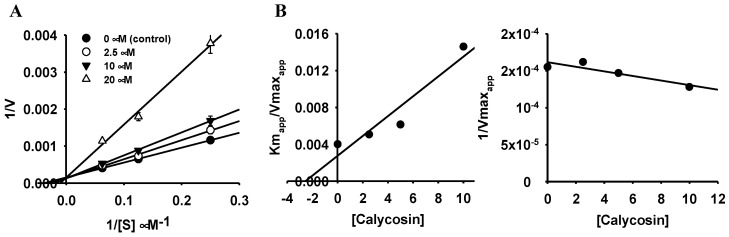
Lineweaver–Burk plots (**A**) and secondary plots (**B**) of calycosin for hMAO-B inhibition.

**Figure 4 ijms-24-15859-f004:**
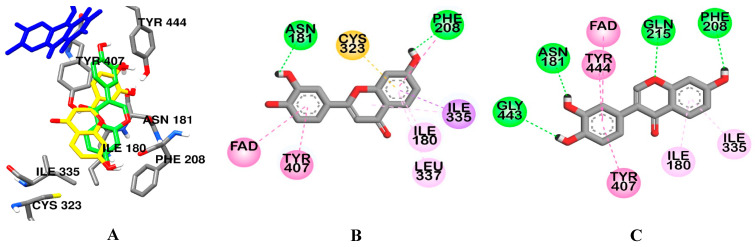
Comparative binding orientation of 3′,4′,7-trihydroxyflavone (yellow stick) and 3′,4′,7-trihydroxyisoflavone (green stick) in the catalytic site of hMAO-A (**A**). FAD and residues of the enzyme are represented in blue and gray sticks, respectively. Binding orientation of 3′,4′,7-trihydroxyflavone (**B**) and 3′,4′,7-trihydroxyisoflavone (**C**) in the catalytic site of hMAO-A.

**Figure 5 ijms-24-15859-f005:**
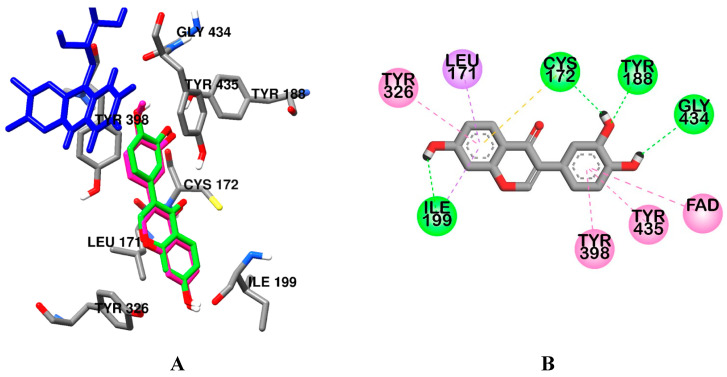
Comparative binding orientation of 3′,4′,7-trihydroxyisoflavone (green stick) in the catalytic site of hMAO-B (**A**). FAD and residues of the enzyme are represented in blue and gray sticks, respectively. Binding orientation of 3′,4′,7-trihydroxyisoflavone (**B**) in the catalytic site of hMAO-B.

**Figure 6 ijms-24-15859-f006:**
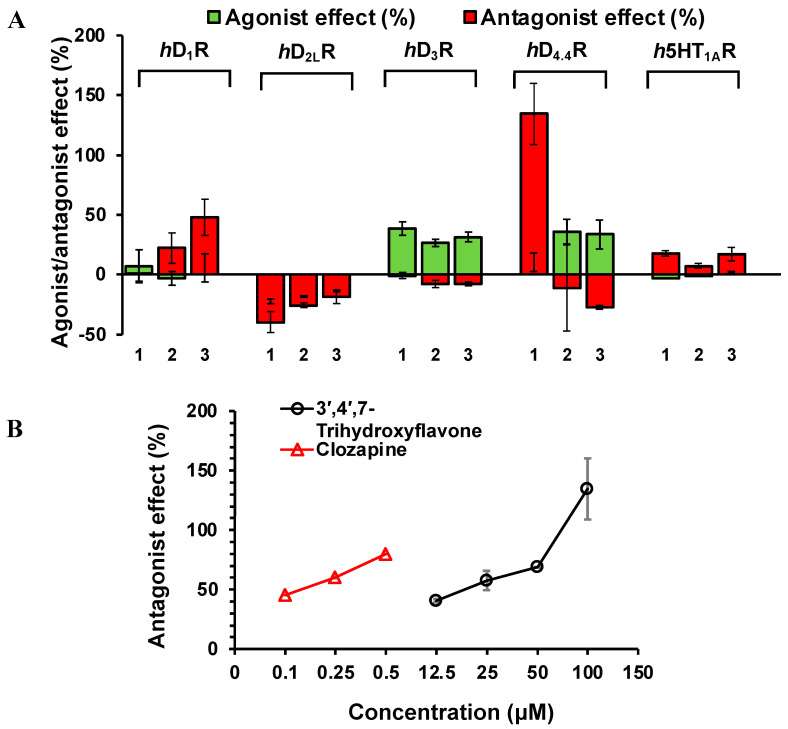
Agonist and antagonist effect (%) of compounds (**1**: 3′,4′,7-trihydroxyflavone, **2**: 3′,4′,7-Trihydroxyisoflavone, and **3**: calycosin) at 100 μM against dopamine D_1_, D_2L_, D_3_, and D_4.4_ receptors and 5HT_1A_ receptors (**A**). Concentration-dependent antagonist effect of 3′,4′,7-trihydroxyflavone (**1**) on dopamine D_4.4_ receptor (**B**).

**Figure 7 ijms-24-15859-f007:**
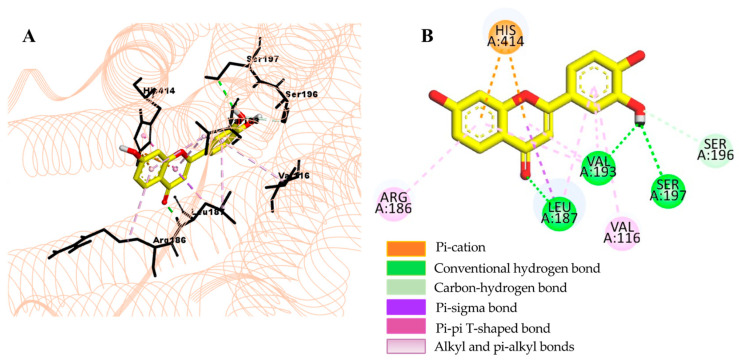
Close-up view of 3′,4′,7-trihydroxyflavone–hD4R complex showing the interactions within the active site of the receptor (**A**). A 2D interaction diagram showing the binding of 3′,4′,7-trihydroxyflavone to the amino acid residues of hD4R (**B**).

**Figure 8 ijms-24-15859-f008:**
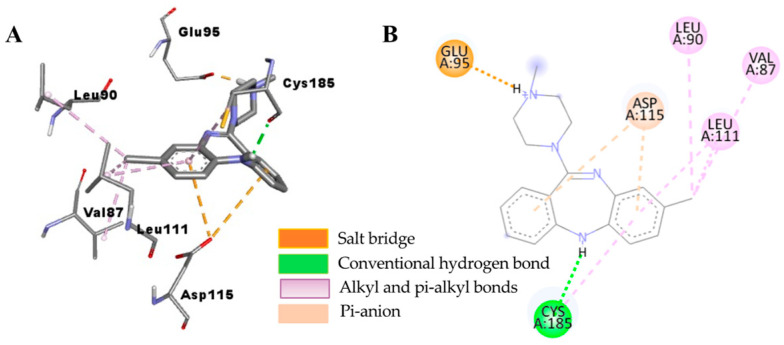
Close-up view of clorgyline–hD4R complex showing the interactions within the active site of the receptor (**A**). A 2D interaction diagram showing the binding of clorgyline to the amino acid residues of hD4R (**B**).

**Figure 9 ijms-24-15859-f009:**
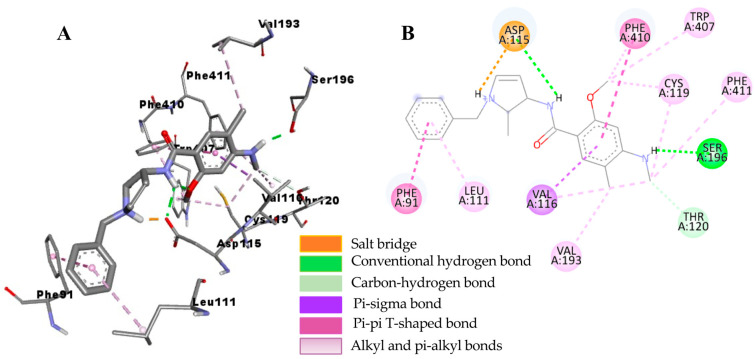
Close-up view of the nemonapride–hD_4_R complex showing the interactions within the active site of the receptor (**A**). A 2D interaction diagram showing the binding of nemonapride to the amino acid residues of hD_4_R (**B**).

**Figure 10 ijms-24-15859-f010:**
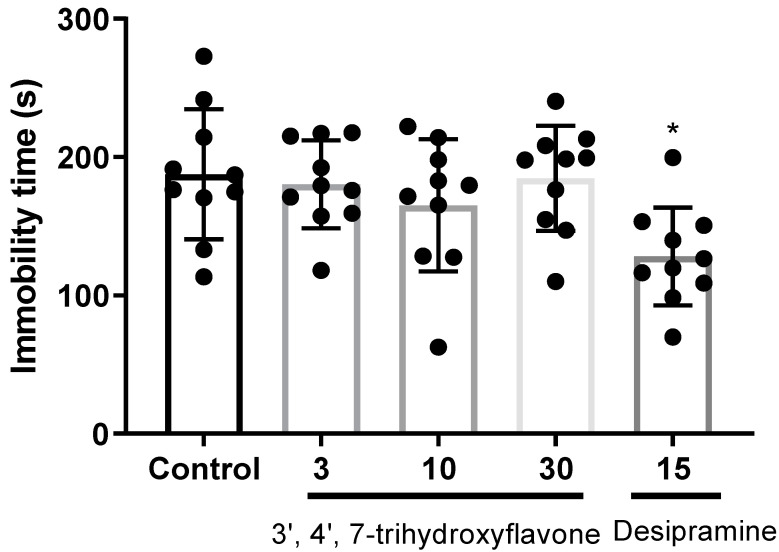
The effects of **1** (3′,4′,7-trihydroxyflavone) on depressive-like behavior were measured with the forced swimming test (FST). FST was performed in a transparent cylindrical glass cylinder with a diameter of 14 cm and a height of 25 cm, filled with 20 cm of water (24 ± 2 °C). Mice were habituated to the experimental environment 1 h before the test. 3′,4′,7-trihydroxyflavone (3, 10, or 30 mg/kg, p.o.) or the same volume of vehicle was administered 1 h before FST. * Statistically significant at a *p*-value < 0.01.

**Table 1 ijms-24-15859-t001:** Recombinant human monoamine oxidase (*h*MAO) inhibitory activity of compounds.

Samples	*h*MAO-A	*h*MAO-B	SI ^b^
	IC_50_ (µM) ^a^		
3′,4′,7-Trihydroxyflavone (**1**)	7.57 ± 0.14	>150	<0.05
3′,4′,7-Trihydroxyisoflavone (**2**)	176.79 ± 7.80	71.23 ± 0.06	2.48
Calycosin (**3**)	113.78 ± 3.39	7.19 ± 0.32	15.82
Deprenyl-HCl ^c^	12.57 ± 0.51	0.38 ± 0.001	33.08
Clorgyline-HCl ^c^	0.02 ± 0.00	NT	NT
Safinamide mesylate ^c^	NT	0.23 ± 0.01	NT
	Inhibition constants (*K_i_*, µM) ^d^		Inhibition mode ^e^
3′,4′,7-Trihydroxyflavone (**1**)	2.03	NT	Competitive
Calycosin (**3**)	NT	2.56	Competitive

^a^ The 50% inhibitory concentration (IC_50_) values (μM) were calculated from a log dose inhibition curve and expressed as mean ± SD of triplicate experiments. ^b^ The selective index (SI) was determined with the ratio of *h*MAO-A IC_50_/*h*MAO-B IC_50._
^c^ Reference control. ^d^ Determined using Lineweaver–Burk plot and its secondary plots. ^e^ Determined secondary plot of slopes (*K*_mapp_/*V*_maxapp_) versus concentrations of inhibitor. NT: Not tested.

**Table 2 ijms-24-15859-t002:** Molecular interaction of hMAO-A (2z5x) active site with active compounds as well as reported inhibitors.

Ligand	Binding Score (kcal/mol)	Interacting Residues ^a^	
		H-Bond	Other Interaction Residues
3′,4′,7-Trihydroxyflavone (**1**)	−8.80	Asn181, Phe208	FAD and Phe208 (Pi–Pi T-shaped), Tyr407 (Pi–Pi Stacked), Ile335 (Pi–Sigma, Pi–Alkyl), Ile180 and Leu337 (Pi–Alkyl), Cys323 (Pi–Sulfur)
3′,4′,7-Trihydroxyisoflavone (**2**)	−8.54	Gln215, Gly443, Asn181, Phe208	FAD (Pi–Pi T-shaped), Tyr407 and Tyr444 (Pi–Pi Stacked), Ile180 and Ile335 (Pi–Alkyl)
Harmine ^b^	−8.43	ND	Tyr407 (Pi–Pi Stacked and Pi–Alkyl), FAD (van der Waals), Cys323 (Pi–Sulfur), Ile335 (Pi–Sigma, Pi–Alkyl), Tyr444, Ile180, and Leu337 (Pi–Alkyl)

^a^ All amino acid residues from the enzyme–inhibitor complex were determined with the AutoDock 4.2 program and Discovery Studio. ^b^ Reported inhibitors. ND, Not detected.

**Table 3 ijms-24-15859-t003:** Molecular interaction of hMAO-B (2v5z) active site with active compounds as well as reported inhibitors.

Ligand	Binding Score (kcal/mol)	Interacting Residues ^a^	
		H-Bond	Other Interaction Residues
3′,4′,7-Trihydroxyisoflavone (**2**)	−9.56	Tyr188, Gly434, Cys172, Ile199	Leu171 and Ile199 (Pi–Sigma), Cys172 (Pi–Sulfur), Tyr398 and Tyr435 (Pi–Pi Stacked), Tyr326 and FAD (Pi–Pi T-shaped)
Calycosin (**3**)	−9.99	Cys172, Ile199	FAD (Pi–Sigma, Pi–Pi T-shaped), Tyr435 (Pi–Sigma and Pi–Pi Stacked), Tyr398 and Tyr326 (Pi–Pi T-shaped), Cys172 (Pi–Sulur), Leu171 and Ile199 (Pi–Sigma)
Safinamide ^b^	−9.23	FAD, Tyr435, Gln206	Cys172 (Pi–Sulfur), Tyr326 (Pi–Pi T-shaped), FAD, Tyr398, Tyr435, Leu171, and Ile199 (Pi–Alkyl)

^a^ All amino acid residues from the enzyme–inhibitor complex were determined with the AutoDock 4.2 program and Discovery Studio. ^b^ Reported inhibitors.

**Table 4 ijms-24-15859-t004:** Efficacy values (% stimulation and % inhibition of control against response) of compounds at dopamine D_1_, D_2L_, D_3_, and D_4.4_ receptors and 5HT_1A_ receptors at 100 μM.

Target GPCRs	% Stimulation ^a^ (% Inhibition ^b^)			EC_50_ ^c^ (IC_50_ ^d^)
	3′,4′,7-Trihydroxyflavone	3′,4′,7-Trihydroxyisoflavone	Calycosin	Positive Control
hD_1_R	6.95 ± 14.07	−3.1 ± 5.66	5.65 ± 11.53	0.027
	(0.7 ± 5.94)	(22.2 ± 13.01)	(47.75 ± 15.06)	(0.00057)
hD_2L_R	−22.45 ± 2.05	−18.35 ± 0.78	−13.8 ± 0.28	0.016
	(−40.05 ± 8.69)	(−25.75 ± 1.91)	(−18.65 ± 5.59)	(0.012)
hD_3_R	38.5 ± 5.94	26.43 ± 2.89	26.43 ± 2.89	0.0045
	(−0.97 ± 2.73)	(−8.0 ± 3.11)	(−7.97 ± 1.80)	(0.031)
hD_4.4_R	10.33 ± 7.46	35.8 ± 10.59	33.63 ± 12.10	0.0072
	(134.47 ± 25.54)	(−11.17 ± 35.99)	(−27.33 ± 1.37)	(0.14)
h5HT_1A_R	−2.9 ± 0.28	−1.1 ± 0.42	2.5 ± 0.42	0.0011
	(18.05 ± 2.33)	(7.3 ± 1.70)	(16.75 ± 5.73)	(0.0015)

^a,b^ % stimulation and % inhibition of control agonist response at 100 μM of compounds, respectively, were calculated from a log dose inhibition curve and expressed as mean ± SD of triplicate experiments. ^c^ EC_50_ (nM) values of standard agonists (D_1_, D_2L_, D_3_, and D_4.4_: dopamine, 5-HT_1A_: serotonin). ^d^ IC_50_ (nM) values of standard antagonists (D_1_: SCH-23390, D_2L_: butaclamol, D_3_: (+)-butaclamol, D_4.4_: clozapine, and 5-HT_1A_: (S)-WAY-100635).

**Table 5 ijms-24-15859-t005:** Molecular interaction of hD4R (5WIU) active site with active compound as well as reported antagonists.

Ligand	Binding Score (kcal/mol)	Interacting Residues ^a^	
		H-Bond	Other Interaction Residues
3′,4′,7-Trihydroxyflavone	−7.71	Val193, Leu187, Ser197, Ser196	His414 (Pi–Pi T-shaped), Leu187 (Pi–sigma), Val193 (Pi–Alkyl), Arg186 and Val116 (Pi–Alkyl)
Clorgyline ^b^ (antagonist)	−8.67	Glu95 (salt bridge), Cys185	Cys185 (Pi–alkyl), Leu111 (Pi–alkyl and Alkyl), Leu90 (Alkyl), VAl87 (Alkyl)
Nemonapride ^b^ (antagonist)	−11.69	Asp115 (salt bridge), Ser196, Thr120	Phe91 (Pi–Pi T-shaped), Asp115 (Pi–Pi T-shaped), Phe410 (Pi–Pi T-shaped, Pi–alkyl), Val116 (Pi–sigma), Leu111 (Pi–Alkyl), Phe411 (Pi–Alkyl), Tyr407 (Pi–Alkyl), Val193 (Alkyl), Cys119 (Alkyl)

^a^ All amino acid residues from the enzyme–inhibitor complex were determined with the AutoDock 4.2 program and Discovery Studio. ^b^ Reported inhibitors.

## Data Availability

All data analyzed during this study are available from the corresponding author upon reasonable request.

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
