# Peer review of "In Vitro Human Monoamine Oxidase Inhibition and Human Dopamine D4 Receptor Antagonist Effect of Natural Flavonoids for Neuroprotection"

_ijms, 2023, doi:10.3390/ijms242115859_

Round 1
Reviewer 1 Report
Comments and Suggestions for Authors
Authors evaluated the inhibitory potential and functional effects of a series of naturally occurring flavonoids at human monoamine oxidase (hMAO) and dopamine + serotonin G protein-coupled receptors (GPCRs), respectively. Inhibitory potency of the compounds in micromolar range was confirmed in molecular docking studies. Finally, authors reported that the hypothesized antidepressant effect of the most potent flavonoid was not detected in the forced swimming test in vivo.
Overall, the study is well-designed and executed. The uncovered antagonistic effect of compound 1 (3,4,7-trihydroxyflavone) at human D4 dopamine receptor is novel. However, authors need to address two important concerns before the manuscript is suitable for publication:
1. The first part of the manuscript (in vitro hMAO inhibition assays) appears to be a replication of a recent study (Calycosin and 8-O-methylretusin isolated from Maackia amurensis as potent and selective reversible inhibitors of human monoamine oxidase-B - PubMed (nih.gov);
Please clarify and incorporate this paper into the References section.
2. The statement in Conclusions that 'the oral route of administration is not effective for the anti-depressant effect of 1 in the male C57BL/6 mice model" appears to be premature. Could authors provide any experimental evidence that altering the administration route produces the hypothesized antidepressant effect?
Author Response
Reviewer 1
Authors evaluated the inhibitory potential and functional effects of a series of naturally occurring flavonoids at human monoamine oxidase (hMAO) and dopamine + serotonin G protein-coupled receptors (GPCRs), respectively. Inhibitory potency of the compounds in micromolar range was confirmed in molecular docking studies. Finally, authors reported that the hypothesized antidepressant effect of the most potent flavonoid was not detected in the forced swimming test in vivo.
Overall, the study is well-designed and executed. The uncovered antagonistic effect of compound 1 (3,4,7-trihydroxyflavone) at human D4 dopamine receptor is novel. However, authors need to address two important concerns before the manuscript is suitable for publication:
Response: All authors are grateful to the reviewer for the thorough review and comments. The manuscript has been revised based on the comments and suggestions.
- The first part of the manuscript (in vitro hMAO inhibition assays) appears to be a replication of a recent study (Calycosin and 8-O-methylretusin isolated from Maackia amurensis as potent and selective reversible inhibitors of human monoamine oxidase-B - PubMed (nih.gov);
Please clarify and incorporate this paper into the References section.
Response: The above-mentioned paper screens hMAO activity of isolated compounds from the stems of Maackia amurensis. In our study, we mainly focused on three flavonoids and evaluated hMAO inhibition activity and correlated to the functional effect on dopamine receptor sub-types. This is not a replication of the above-mentioned paper being calycosin a common test compound. We have cited this article following its discussion in the Discussion section.
- The statement in Conclusions that 'the oral route of administration is not effective for the anti-depressant effect of 1 in the male C57BL/6 mice model" appears to be premature. Could authors provide any experimental evidence that altering the administration route produces the hypothesized antidepressant effect?
Response: We included an explanation along with the reference in the Conclusion section.
A recent study confirmed that intravenous injection of cannabidiol can induce long-term antidepressant-like effects [1]. The findings indicated that 10 mg/kg cannabidiol (i.v.) or 100 mg/kg CBD (p.o.) administered weekly for four weeks can significantly improve depressive symptoms. In contrast, no significant effects were observed at 10 mg/kg cannabidiol p.o. In our study, oral administration of 1 in the male C57BL/6 mice model at a dose of up to 30 mg/kg did not show an antidepressant effect. So, the effect should be evaluated from different routes of administration or at higher doses following p.o. administration.
Reference
- Xu, C.; Chang, T.; Du, Y.; Yu, C.; Tan, X.; Li, X., Pharmacokinetics of oral and intravenous cannabidiol and its antidepressant-like effects in chronic mild stress mouse model. Environmental toxicology and pharmacology 2019, 70, 103202.
In addition to addressing these comments, we have revised the manuscript based on other reviewer’s comments as well. Kindly, see the revised file and write back to us. We appreciate your feedback.

Reviewer 2 Report
Comments and Suggestions for Authors
This paper titled ‘In Vitro Human Monoamine Oxidase Inhibition and Human Dopamine D4 Receptor Antagonist Effect of Natural Flavonoids for Neuroprotection’ discusses the prevalence of neurological disorders in the United States, which affect more than 200 million people, or roughly 60% of the population. These disorders range in severity from tension-type headaches and anxiety to more serious conditions like strokes and dementia. Notable statistics from the paper include more than 3.4 million people with epilepsy, 6.2 million with Alzheimer's or related dementias, and 40 million with anxiety disorders. Major depressive disorder is highlighted as the leading cause of disability among people aged 15 to 44, with 19% of adults experiencing mental illness in 2019.
The paper emphasizes that the causes of these disorders vary, and in some cases, they are not completely understood due to multifactorial factors, making treatment challenging. The rates of people seeking treatment for mental health issues have also increased significantly, as indicated by the rise in the number of screening tests for anxiety disorders and depression between 2019 and 2020.
The paper points out the urgent need for novel therapeutic strategies to manage these neurological disorders. It discusses different classes of antidepressant drugs available in the market and their mechanisms of action, highlighting their associated side effects. The authors mention that natural flavonoids derived from sources like soybeans and medicinal plants have shown promising therapeutic effects against various medical conditions, including neurological disorders, cancer, inflammation, and depression.
The paper delves into the potential of three naturally occurring flavonoids: 3′,4′,7-trihydroxyflavone, 3′,4′,7-trihydroxyisoflavone, and calycosin, for inhibiting human monoamine oxidases (hMAOs) and modulating G protein-coupled receptors (GPCRs). In vitro experiments showed these flavonoids' inhibitory potential against hMAO-A and hMAO-B, which are implicated in various neurological conditions. The authors conducted molecular docking simulations to gain insights into the interactions between these flavonoids and the hMAOs, further confirming their inhibitory potential.
In summary, this paper highlights the prevalence of neurological disorders in the United States and explores the potential of natural flavonoids, particularly 3′,4′,7-trihydroxyflavone, for their inhibitory effects on human monoamine oxidases and modulation of G protein-coupled receptors as a possible avenue for developing novel therapeutic strategies for managing these disorders.
In general, I think the idea of this article is really interesting and the authors’ fascinating observations on this timely topic may be of interest to the readers of IJMS. However, some comments, as well as some crucial evidence that should be included to support the author’s argumentation, needed to be addressed to improve the quality of the manuscript, its adequacy, and its readability prior to the publication in the present form. My overall judgment is to publish this paper after the authors have carefully considered my suggestions below, in particular reshaping parts of the ‘Introduction’ and ‘Methods’ sections by adding more evidence.
Please consider the following comments:
- The introduction of the paper should be revised to provide a clearer overview of the paper's scope and objectives. Currently, the introduction lacks a clear structure and a statement of the research problem or hypothesis. Consider restructuring it to better introduce the context and importance of the research.
- The introduction provides an adequate overview of the prevalence of neurological disorders and the need for novel therapeutic strategies. However, it could benefit from more recent statistics or references, as the most recent data mentioned is from 2019. In this regard, consider expanding the introduction to include a brief discussion of the neural substrates involved in the pathophysiology of neurological disorders. Providing a concise overview of the key neural substrates or pathways that play a critical role in these disorders could help set the stage for the relevance of the study's focus on monoamine oxidase inhibitors and flavonoids as potential therapeutic strategies. This additional context will help readers better understand the underlying neural mechanisms and why the study's findings are significant in the broader context of neurological research [1-4].
- Interpretation of Results: The discussion of the results should be expanded to provide a more comprehensive interpretation. Explain the potential implications of the IC50 values for different compounds and their selectivity for hMAO-A and hMAO-B. Discuss how these findings relate to the study's objectives and potential applications in the field.
- The discussion provides a good context for the results obtained. It explains the significance of the research in relation to monoamine oxidases, neurotransmitters, and potential applications for various neurological conditions.
- The limitations and future directions section effectively points out the limitations of the study and suggests areas for further research.
- The paper should be reviewed for formatting, including the proper inclusion of figures and tables, and for improved clarity and readability.
- It's essential to provide references to support the claims made in the introduction and discussion sections.
- Consider providing more details about the research methodology and the specific methods used in the experiments.
- Ensure that the research paper follows a clear and organized structure, with appropriate headings and subheadings.
References:
1. https://doi.org/10.3389/fnmol.2023.1217090
2. https://doi.org/10.3390/biomedicines11051248
3. DOI: 10.3390/biomedicines11030945
4. https://doi.org/10.3389/fpsyt.2023.1264669
Comments on the Quality of English Language
Minor english editing is required.
Author Response
Reviewer 2
This paper titled ‘In Vitro Human Monoamine Oxidase Inhibition and Human Dopamine D4 Receptor Antagonist Effect of Natural Flavonoids for Neuroprotection’ discusses the prevalence of neurological disorders in the United States, which affect more than 200 million people, or roughly 60% of the population. These disorders range in severity from tension-type headaches and anxiety to more serious conditions like strokes and dementia. Notable statistics from the paper include more than 3.4 million people with epilepsy, 6.2 million with Alzheimer's or related dementias, and 40 million with anxiety disorders. Major depressive disorder is highlighted as the leading cause of disability among people aged 15 to 44, with 19% of adults experiencing mental illness in 2019.
The paper emphasizes that the causes of these disorders vary, and in some cases, they are not completely understood due to multifactorial factors, making treatment challenging. The rates of people seeking treatment for mental health issues have also increased significantly, as indicated by the rise in the number of screening tests for anxiety disorders and depression between 2019 and 2020.
The paper points out the urgent need for novel therapeutic strategies to manage these neurological disorders. It discusses different classes of antidepressant drugs available in the market and their mechanisms of action, highlighting their associated side effects. The authors mention that natural flavonoids derived from sources like soybeans and medicinal plants have shown promising therapeutic effects against various medical conditions, including neurological disorders, cancer, inflammation, and depression.
The paper delves into the potential of three naturally occurring flavonoids: 3′,4′,7-trihydroxyflavone, 3′,4′,7-trihydroxyisoflavone, and calycosin, for inhibiting human monoamine oxidases (hMAOs) and modulating G protein-coupled receptors (GPCRs). In vitro experiments showed these flavonoids' inhibitory potential against hMAO-A and hMAO-B, which are implicated in various neurological conditions. The authors conducted molecular docking simulations to gain insights into the interactions between these flavonoids and the hMAOs, further confirming their inhibitory potential.
In summary, this paper highlights the prevalence of neurological disorders in the United States and explores the potential of natural flavonoids, particularly 3′,4′,7-trihydroxyflavone, for their inhibitory effects on human monoamine oxidases and modulation of G protein-coupled receptors as a possible avenue for developing novel therapeutic strategies for managing these disorders.
In general, I think the idea of this article is really interesting and the authors’ fascinating observations on this timely topic may be of interest to the readers of IJMS. However, some comments, as well as some crucial evidence that should be included to support the author’s argumentation, needed to be addressed to improve the quality of the manuscript, its adequacy, and its readability prior to the publication in the present form. My overall judgment is to publish this paper after the authors have carefully considered my suggestions below, in particular reshaping parts of the ‘Introduction’ and ‘Methods’ sections by adding more evidence.
Response: Thank you very much for a thorough review of our article and encouraging comments. We are grateful to have such constructive comments. We have revised the manuscript based on the reviewer’s comments and suggestions.
Please consider the following comments:
- The introduction of the paper should be revised to provide a clearer overview of the paper's scope and objectives. Currently, the introduction lacks a clear structure and a statement of the research problem or hypothesis. Consider restructuring it to better introduce the context and importance of the research.
Response: We have revised the Introduction part as below.
Modern research on natural flavonoids has revealed that dietary consumption of flavonoids and flavonoids-rich foods significantly improve cognition and delay age-related neurodegenerative disorders via inhibition of cholinesterase and β-secretase (BACE1), antioxidant mechanism, and modulation of signaling pathways, that are implicated in cognitive and neuroprotective functions [27, 28]. What remains lacking, however, is knowledge of the potential role of natural flavonoids in regulating aminergic pathways. There is, therefore, a critical need to define the therapeutic efficacy of natural flavonoids in depression-like animal models. Without such information, the promise of natural flavonoids for the treatment of neurodegenerative diseases will likely remain limited.
Our long-term goal is to discover natural flavonoids as hMAO inhibitors and hD4R antagonists for the management of neurodegenerative diseases, particularly depression. Our overall objectives, which are the next step toward the attainment of our long-term goal, are to (i) elucidate the mechanism(s) of hMAO inhibition via in vitro enzyme assays and in silico molecular docking, (ii) evaluate modulating effect on dopamine (DA) and serotonin (5-HT) receptors via GPCR-functional assays, and in silico molecular docking, and (iii) determine their in vivo anti-depressant efficacy using depression-like animal models. Our central hypothesis is that natural flavonoids with in vitro hMAO inhibition and hD4R antagonism have in vivo anti-depressant effects. Therefore, to attain the overall objectives and test the hypothesis, we emphasized the “One-compound multiple-targets paradigm” and evaluated the role of three naturally occurring flavonoids, namely, 3′,4′,7-trihydroxyflavone (1), 3′,4′,7-trihydroxyisoflavone (2), and calycosin (3) in hMAO inhibition and GPCRs modulation followed by in vivo anti-depressant effect in depression-like animal models.
- The introduction provides an adequate overview of the prevalence of neurological disorders and the need for novel therapeutic strategies. However, it could benefit from more recent statistics or references, as the most recent data mentioned is from 2019. In this regard, consider expanding the introduction to include a brief discussion of the neural substrates involved in the pathophysiology of neurological disorders. Providing a concise overview of the key neural substrates or pathways that play a critical role in these disorders could help set the stage for the relevance of the study's focus on monoamine oxidase inhibitors and flavonoids as potential therapeutic strategies. This additional context will help readers better understand the underlying neural mechanisms and why the study's findings are significant in the broader context of neurological research [1-4].
Response:
Tryptophan (TRP) is the most prevalent amino acid, which plays a significant role in protein biosynthesis, and its metabolites are implicated in central nervous system (CNS) disorders [1, 2]. This amino acid metabolizes to 5-hydroxytryptophan (5-HT) and kynurenine (Kyn) via two pathways: the serotonin and kynurenine pathways, respectively [3, 4]. Tryptophan hydroxylase 1 or 2 converts tryptophan to 5-hydroxytryptophan (5-HTP), the first rate-limiting step in the 5-HT pathway. 5-HTP is then decarboxylated by aromatic acid decarboxylase (AADC) to form 5-HT, which is metabolized by aralkylamine N-acetyltransferase (AANAT) to N-acetyl serotonin (NAS), and then by N-acetylserotonin O-methyltransferase (ASMT) to form melatonin. 5-HT is also metabolized by monoamine oxidase (MAO) to form 5-hydroxyindoleacetic acid (5-HIAA), which is the main metabolite of 5-HT [5]. Similarly, Trp is converted to kynurenine (Kyn) via indoleamine 2-3-dioxygenases 1 and 2 and tryptophan 2,3-dioxygenase of the Kyn pathway. Kyn is then converted to 3-hydroxyKyn (3-HK) via kynurenine 3-monooxygenase. Kynurenine aminotransferase (KAT) then converts 3-HK to xanthurenic acid (XA), which is converted to 3-hydroxyanthranilic acid (3-HAA) by kynureninase (Kynu). 3-HAA is metabolized to picolinic acid (PA) via aminocarboxymuconate-semialdehyde decarboxylase (ACMSD) and nonenzymatically to quinolinic acid (QA). QA is then metabolized to NAD+ by quinolinate phosphoribosyl transferase (QPRT) [6]. These tryptophan metabolites play important roles in many psychiatric disorders, including depression [7, 8].
TRP and its metabolites play an important role in alleviating various diseases ranging from psychiatric/neurological disorders to cancer [9]. This distinctive feature makes the study of TRP metabolism an exciting area of ​​research for biomedical researchers focused on developing and identifying new therapeutic targets. Recent research suggests that the Kynurenine pathway can improve many biological systems that function poorly in psychiatric disorders, including the neurotransmitters of the CNS and the immune-inflammatory system [10, 11]. The Kynurenine pathway event-targeted process represents an excellent opportunity to develop effective treatments for neuropsychiatric disorders [1, 12].
- Interpretation of Results: The discussion of the results should be expanded to provide a more comprehensive interpretation. Explain the potential implications of the IC50 values for different compounds and their selectivity for hMAO-A and hMAO-B. Discuss how these findings relate to the study's objectives and potential applications in the field.
Response: We have revised our Result and Discussion section as needed focusing on IC50 values, selectivity, and structural activity relationship. Please see the revised file.
- The discussion provides a good context for the results obtained. It explains the significance of the research in relation to monoamine oxidases, neurotransmitters, and potential applications for various neurological conditions.
- The limitations and future directions section effectively points out the limitations of the study and suggests areas for further research.
- The paper should be reviewed for formatting, including the proper inclusion of figures and tables, and for improved clarity and readability.
Response: We reviewed the paper for formatting, including the proper inclusion of figures and tables, and for improved clarity and readability and revised as necessary.
- It's essential to provide references to support the claims made in the introduction and discussion sections.
Response: As mentioned above, we have revised the Introduction section and included the latest relevant references.
- Consider providing more details about the research methodology and the specific methods used in the experiments.
Response: All the research methodologies followed here are reported in detail in our recent publications in IJMS. Still, we have tried our best to explain in detail.
- Ensure that the research paper follows a clear and organized structure, with appropriate headings and subheadings.
Response: We have ensured that the research paper follows a clear and organized structure, with appropriate headings and subheadings.
References:
- https://doi.org/10.3389/fnmol.2023.1217090
- https://doi.org/10.3390/biomedicines11051248
- DOI: 10.3390/biomedicines11030945
- https://doi.org/10.3389/fpsyt.2023.1264669
Response: We have discussed these articles and cited them accordingly. Please refer to the revised file.
References
- Davidson, M.; Rashidi, N.; Nurgali, K.; Apostolopoulos, V., The role of tryptophan metabolites in neuropsychiatric disorders. International journal of molecular sciences 2022, 23, (17), 9968.
- Comai, S.; Bertazzo, A.; Brughera, M.; Crotti, S., Tryptophan in health and disease. Advances in clinical chemistry 2020, 95, 165-218.
- Li, D.; Yu, S.; Long, Y.; Shi, A.; Deng, J.; Ma, Y.; Wen, J.; Li, X.; Liu, S.; Zhang, Y., Tryptophan metabolism: Mechanism-oriented therapy for neurological and psychiatric disorders. Frontiers in Immunology 2022, 13, 985378.
- Polyák, H.; Galla, Z.; Nánási, N.; Cseh, E. K.; Rajda, C.; Veres, G.; Spekker, E.; Szabó, Á.; Klivényi, P.; Tanaka, M., The tryptophan-kynurenine metabolic system is suppressed in cuprizone-induced model of demyelination simulating progressive multiple sclerosis. Biomedicines 2023, 11, (3), 945.
- Höglund, E.; Øverli, Ø.; Winberg, S., Tryptophan metabolic pathways and brain serotonergic activity: a comparative review. Frontiers in endocrinology 2019, 158.
- Savitz, J., The kynurenine pathway: a finger in every pie. Molecular psychiatry 2020, 25, (1), 131-147.
- Maffei, M. E., 5-Hydroxytryptophan (5-HTP): Natural occurrence, analysis, biosynthesis, biotechnology, physiology and toxicology. International journal of molecular sciences 2020, 22, (1), 181.
- Battaglia, M. R.; Di Fazio, C.; Battaglia, S., Activated tryptophan-kynurenine metabolic system in the human brain is associated with learned fear. Frontiers in Molecular Neuroscience 2023, 16, 1217090.
- Keegan, M. R.; Chittiprol, S.; Letendre, S. L.; Winston, A.; Fuchs, D.; Boasso, A.; Iudicello, J.; Ellis, R. J., Tryptophan metabolism and its relationship with depression and cognitive impairment among HIV-infected individuals. International Journal of Tryptophan Research 2016, 9, IJTR. S36464.
- Muneer, A., Kynurenine pathway of tryptophan metabolism in neuropsychiatric disorders: pathophysiologic and therapeutic considerations. Clinical Psychopharmacology and Neuroscience 2020, 18, (4), 507.
- Badawy, A. A.-B., Kynurenine pathway and human systems. Experimental Gerontology 2020, 129, 110770.
- Benedetti, F.; Aggio, V.; Pratesi, M. L.; Greco, G.; Furlan, R., Neuroinflammation in bipolar depression. Frontiers in Psychiatry 2020, 11, 71.

Round 2
Reviewer 2 Report
Comments and Suggestions for Authors
Dear Authors,
I am pleased to acknowledge that you have indeed addressed all of my concerns and queries in a clear and precise manner. Your responses have provided valuable insights into the modifications made to the manuscript in light of my comments. It is evident that you have taken great care to ensure that the revised manuscript aligns more closely with the scientific rigor expected for publication in IJMS. Having reviewed the revised manuscript, I am satisfied with the changes that have been implemented.
In light of the above, I am pleased to recommend acceptance of your manuscript for publication in IJMS.
Best regards,
Reviewer